# Implementation of a Standard Care Program of Therapeutic Exercise in Metastatic Breast Cancer Patients

**DOI:** 10.3390/ijerph191811203

**Published:** 2022-09-06

**Authors:** Bella Pajares, Cristina Roldán-Jiménez, Emilio Alba, Antonio I. Cuesta-Vargas

**Affiliations:** 1UGCI Oncología Médica Hospitales Universitarios Regional y Virgen de la Victoria, 29010 Málaga, Spain; 2Departamento de Fisioterapia, Facultad de Ciencias de la Salud, Universidad de Málaga, Andalucia Tech, 29071 Málaga, Spain; 3Instituto de Investigación Biomédica de Málaga (IBIMA), 29590 Málaga, Spain; 4School of Clinical Science, Faculty of Health Science, Queensland University Technology, 2 George St., Brisbane City, QLD 4000, Australia

**Keywords:** breast cancer, metastasis breast cancer, community, exercise therapy, rehabilitation, palliative care, physiotherapy

## Abstract

Background: There is little information on the feasibility and benefit of therapeutic exercise (TE) in women with metastatic breast cancer (MBC). The aim of this article is to describe the implementation of a TE intervention in MBC patients, and to determine the recruitment, compliance and improvement in outcomes after its completion. Methods: The “Therapeutic Exercise program in MBC” (TEP-MBC) consists of 1 h of individualized TE supervised by a physiotherapist in a group format, consisting of four groups of seven to eight participants. TEP-MBC was delivered twice a week, lasting 12 weeks (22 sessions), with patients considered to have completed the program when attending at least 17 sessions (>75% attendance). After referral, patients underwent a clinical interview and a physical and functional assessment. This information was complemented with patient-reported outcomes. Data about referral, compliance and assessment were collected. Results: Only 11 of the 30 patients completed the program. Drop-out was mainly related to personal issues and symptoms arising from the disease or treatment. All patients who completed the program improved cancer-related fatigue and increased their functional parameters. Conclusions: The TEP-MBC was safe and feasible in patients with MBC, although with low compliance. The high variability in baseline measures reflects the heterogeneous level of function.

## 1. Introduction

Physical activity (PA) displays great benefits in breast cancer (BC) patients and survivors. PA levels are associated with a lower risk of relapse [1], greater BC-specific survival and greater overall survival independent of body mass index or menopausal status [2,3]. PA displays multiple mediated pathways such as metabolism, inflammatory and immunomodulatory effects [4], and essentially lowers all obesity-related mediators of cancer [5]. Besides these benefits, BC patients decrease their PA levels after surgical treatment [6].

Exercise is a planned, structured and repetitive form of physical activity aimed at improving a specific physical benefit [7]. There has been growing interest in therapeutic exercise (TE). In BC patients, TE interventions improve quality of life (QoL) [8], mobility [9] and cancer-related fatigue (CRF). The safety, feasibility and benefit of TE is robust in BC and advanced cancer patients [10]. However, there is little information on the feasibility and benefit of exercise in women with metastatic breast cancer (MBC). 

A prospective randomized study with 100 patients with MBC showed no impact on functionality after an unsupervised distance-based PA program vs. the wait-list control group [11]. Another distance-based study evaluated the impact of a seated exercise DVD program in 38 MBC patients, reporting an improvement in QOL compared to control participants [12]. The literature is limited about the benefit of survival. Analysis of secondary data with more than 100 MBC patients showed that greater PA level at baseline was significantly associated with longer survival, with the major limitation of relying on self-reported questionnaires to draw these conclusions [13]. 

New treatments allow MBC patients to live for several years after diagnosis of metastatic disease. However, many women experience significant side effects from systemic treatments and cancer symptoms in this period, hindering and preventing PA programs and lowering their functional capacity [14]. In this regard, data from a systematic review and meta-analysis have shown that distance-based interventions have a very small, limited impact on PA behavior in BC survivors [15]. Therefore, new approaches are needed to facilitate and support PA levels in MBC patients to achieve more reliable, accurate information on its impact on functionality, quality of life and prognosis.

Current exercise therapy prescription in the oncology setting is limited to generic guidelines, and there is interest in tailoring TE interventions thanks to its effectiveness [7]. Another limitation of exercise therapy prescription is implementation, as there is a lack of accessible exercise therapy prescription in cancer patients in the real-world setting [16]. Some institutions, such as the Canadian Cancer Society [17] and the Australian public hospitals [18] have developed real-world settings. Still, current research is limited to cancer survivors or people diagnosed with cancer [19,20]. The aim of this article was (a) to describe the process behind a free, not-for-profit community-based therapeutic exercise program (TEP) for MBC patients in the clinical setting and (b) to determine the recruitment, compliance and improvement in outcomes after its completion.

## 2. Materials and Methods

### 2.1. Program Design and Description 

The present program started in May 2017 as The School of Healthy Habits for Women Operated for Breast Cancer, known colloquially as The Onco-Health Club (OHC), providing TE and educational interventions in BCS that have been surgically treated for their primary tumor with no evidence presence of tumor or metastatic disease [21]. Later, TE intervention was offered to patients diagnosed with MBC: The Therapeutic Exercise program in MBC (TEP-MBC). 

TEP-MBC was a result of the research network between the Translation Research in Cancer B-01 and Clinimetric F-14 research groups at Málaga Biomedical Research Institute (IBIMA), accredited for healthcare research in Spain by Carlos III Institute of Health (www.ibima.eu/en). The main goal of TEP-MBC was to provide MBC patients the opportunity to benefit from an exercise tailored to their needs that was supervised by a physiotherapist (CRJ). 

### 2.2. Participant Referral and Eligibility

Women were recruited by Medical Oncologists from the Medical Oncology Unit at the hospital (BCP, EA), who were in close contact with physiotherapists (CRJ, ACV). Participants included in this study were diagnosed with metastatic breast cancer, not amenable to curative treatment. Patients were excluded if they had suffered any cardiovascular event defined as stable or unstable angor, acute pulmonary edema, cardiac rhythm disorders or syncope of cause not affiliated in the year prior to inclusion. 

At the beginning of the program, a baseline assessment was carried out.

### 2.3. Clinical Data Collection

Once eligibility was confirmed, all participants in this study signed an informed consent form prior to inclusion. The University Clinical Hospital gave ethical clearance for the study, following the Declaration of Helsinki. The oncologists collected clinical data on tumor subtype and type of surgery (breast-conserving or mastectomy), line of treatment, type of ongoing systemic therapy (endocrine therapy (ET), ET-cyclin-dependent kinases (CDKs) inhibitors, chemotherapy (CT), monoclonal antibody (MA) and CT-MA, type of metastatic disease (oligometastatic or multiple metastasis) and location of metastatic disease (visceral, not visceral, or both). The presence of bone metastases, axial bone metastasis, spine stabilization surgery and type and metastatic bone pattern (osteolytic, osteoblast or mixed) were also collected. This allowed screening precautions, assessing possible risk and tailoring intervention. For example, modifications in cases of bone metastases based on location [22,23].

### 2.4. Clinical Interview

Patients underwent an interview with the physiotherapist (CRJ), reporting their clinical history, ensuring personalized intervention based on the clinical information, the current interview, and further physical testing to establish baseline levels [22,24]. Patients were given personalized information about ET benefits [25], and any questions were answered, e.g., the effects of lifting weights [26] and the safety of upper limb strength exercises on lymphedema [27]. This allowed patients’ interests and preferences to be considered [28]. Furthermore, patients were asked about prior or current exercise behavior to establish intensity levels [24]. 

### 2.5. Physical and Functional Assessment

Assessment consisted of assessing the musculoskeletal system, the cardiorespiratory capacity and muscular strength. The physical assessment allowed musculoskeletal signs and symptoms [24], range of motion limitations and motor control to be taken into account to establish adaptations, loads and targeted muscle groups. For example, the range of motion in upper limb exercises was modified in cases of pectoral shortening and/or skin retraction on the affected side. To assess cardiorespiratory capacity, a submaximal oncology ergometry was carried out following a protocol tested in BC survivors [29]. It consisted of a multistage treadmill test, increasing speed gradually until the patients reached 85% of their maximum predicted heart rate (HR). According to the literature, muscular strength was assessed in the major muscle groups [30]. The program’s weight was based on the estimated percentage of one-repetition maximum (1-RM). A weight load that produces fatigue in the 12 repetitions (12-RM) was calculated for better strength gains [31,32].

Besides the widely-known 30-Second sit-to-stand test (30-STS) [33] and handgrip strength test [34], the functional assessment was provided by the following test: Lie-to-sit (LTS) transfer: Patients were asked to transfer from lying to sitting. Patients started from a supine position with the head resting and arms parallel to the body. The patient should turn right, supporting the right arm to arise from the sitting position. They were allowed to use a hand and a pillow if necessary. The number of repetitions performed during 30 s was counted [35]Adapted burpees (AB): Furthermore, patients who were able to complete the 30-STS test with repetitions ≥15 and BPE ≤ 7 (strong) were asked to perform adapted burpees (AB) for two minutes, following a protocol tested in BCS [29].

This information was complemented with the following questionnaires: Piper Fatigue Scale-Revised (PFS-R) [36] to measure cancer-related fatigue (CRF), the Upper Limb Functional Index (ULFI) [37], the Lower Limb Functional Index (LLFI) [38], the International Physical Activity Questionnaire–Short Form (IPAQ–SF) [39], the European Organization for Research and Treatment of Cancer Quality of Life Questionnaire Core 30 (EORTC QLQ–C30) [40] and the European Organization for Research and Treatment of Cancer Breast Cancer-Specific Quality of Life Questionnaire (EORTC QLQ–BR23) [41]. 

### 2.6. Therapeutic Exercise Intervention

The intervention consisted of 1 h of individualized TE supervised by a physiotherapist. It was delivered twice a week, lasting 12 weeks. The intervention was in a group format, consisting of 4 groups of 7–8 participants. Thus, the complete program consisted of 22 TE sessions, with patients being taken to have completed the program when attending at least 17 sessions (>75% attendance). It was considered that patients who had missed more than five sessions did not complete the program.

Exercises to generate neuromuscular and cardiovascular adaptations were carried out, accounting for training principles [7] and current recommendations in the oncology field [9]. Intensity, time and time prescription were individualized based on evaluations of muscular strength, endurance and patient needs [24], and followed the FIIT formula, as detailed in Appendix A. The TE intervention consisted mainly of strength exercises and endurance with aerobic training.

### 2.7. Funding and Sustainability

Contract Nº PS16060 in IBIMA between Novartis-IBIMA funded the TEP-MBC, consisting of payment for CRJ as the physiotherapist. University Clinical Hospital Virgen de la Victoria provided the rehabilitation room, equipped with bicycles, dumbbells, weights, mats and treatment tables for those patients unable to lay on the floor. The Chair of Physiotherapy at the University of Málaga provided material for assessment.

### 2.8. Statistical Analysis

Descriptive analyses were used to present the mean and standard deviation of quantitative variables and number (percentage) for qualitative variables. Patients’ functional and self-reported outcome data were calculated at baseline and post-intervention. The following groups were considered: baseline group (all patients who attended the assessment, left or not started group, compliance < 75% group and intervention group (Compliance > 75%). Mean changes were calculated in the intervention group. All analyses were performed using SPSS 22.0 for Windows.

## 3. Results

### 3.1. Recruitment 

A total of 30 women who were MBC patients were recruited as volunteers between February 2018 and April 2019 by Medical Oncologists from the Medical Oncology Unit at University Clinical Hospital Virgen de la Victoria (Málaga, Spain). 

### 3.2. Compliance 

Of the 30 patients initially recruited, only 11 completed the program with attendance at 17 sessions or more (75% of attendance). In total, 19 patients abandoned the program due to different reasons: 11 patients dropped out because of personal matters (transport, family problems, distance to hospital, lack of motivation); 5 patients presented tumor symptoms (3 bone pain, 1 liver pain, and 1 thrombosis) that prevented them from attending the sessions; 2 patients had serious treatment side effects; and 1 patient presented major depressive symptoms that made it impossible to complete the program.

Overall, 11 patients completed the program, representing 36% compliance. Lack of attendance was due to personal matters and health status, namely hematological compromise (absolute neutrophil counts), pharyngitis and wound infection. In total, 4 out of 11 patients lacked some functional parameters because they were unable to attend on the day of the pre- or post-evaluation and seven out of 11 patients completed the program having attended all the evaluation appointments (pre- and post-intervention), meaning complete data are available.

### 3.3. Clinical and Oncology Variables

The average age of participants was 52.46 years. Affected breast side and comorbidities from the baseline and intervention group are shown in Table 1.

Regarding treatment (*n* = 30), the majority presented Hormone Receptor (HR) positive—HER2 negative tumors (21 patients, 70%) and were treated in the 1st line of treatment (20 patients, 67%). Three patients presented HR negative—HER2 positive tumors, three presented HR positive—HER 2 positive tumors and three presented triple negative (TN) tumors. Six patients were treated in the 2nd line and four in the 3rd line of treatment. More details from the baseline and intervention group are given in Table 2.

### 3.4. Patient-Reported and Functional Outcomes at Baseline

Baseline levels from the whole sample are provided in Table 3. As can be observed, the group who left or did not start the program had lower levels of physical activity measured by IPAQ (2.351 METS), and slightly lower levels of function measured by 30-STS and LLFI. 

### 3.5. Improvements in Outcomes

Differences between baseline and final assessment were calculated in those patients with Compliance > 75% (intervention group). Regarding CRF, the intervention group showed lower levels after intervention (4.33). With regards to QoL, results varied depending on the questionnaire used. All patients who completed the program objectively increased their functional parameters concerning 30-STS and LTS. In the intervention group, 30-STS increased from 14.50 repetitions to 19.61 repetitions. Regarding lie-to-sit (LTS) transfer, data could only be attained pre- and post-intervention from 6 of the 11 patients, and all of them showed an increase in their scores. Regarding handgrip strength, values remained stable. More details are shown in Table 4.

## 4. Discussion

As far as the authors are aware, this is the first study that provides a supervised TE program in MBC patients, which has shown to be safe and feasible in our study population, with no adverse events related to exercise reported in the group. Moreover, this study provides the descriptive functional status of MBC patients based on patient-reported outcomes and functional assessment, as well as data in terms of recruitment, compliance and improvement in outcomes after its completion. It should be noted that, while exercise in BCS under adjuvant therapy and beyond is widely studied, this study is the second with a supervised design in metastatic cancer patients [42], and the first in a homogeneous cohort of MBC patients. Therefore, the results of this study may be of interest to clinicians and researchers when implementing and designing TEP in MBC patients in a real-world setting.

### 4.1. Recruitment

One strength of the present study was that oncologists recruited patients interested in the intervention. Current guidelines highlight that oncologists’ advice and referral to exercise programs are essential for patient engagement [43]. Furthermore, oncologists provided oncology and clinical data to the physiotherapist, guaranteeing screening precautions, such as bone metastases (Table 2).

### 4.2. Compliance 

Drop-out was the main obstacle in developing our TE program, with only 11 out of 30 patients recruited completing the program, representing 36% compliance. Most patients who dropped out did so for reasons unrelated to disease progression or treatment side effects. This is essential, as most women who dropped out did so mainly due to reasons unrelated to exercise (Figure 1).

Similar interventions in BCS have shown compliance of 67%, and drop-outs and lack of attendance were mainly related to personal matters (incompatibility with work and family life), health problems and transport barriers [21]. Data from other randomized studies in metastatic patients report higher rates of completion (64–70%) [11,42], but recent reports from other hospital distance-based PA programs prescribed at home are around 35–52% [44], with abandonment due mainly to tumor progression [42]. The reason for our low compliance rate may be the cohort heterogeneity, with significant differences in extension of metastatic disease and line of treatment. Although compliance in MBC population is still known, it should be highlighted that this population suffers from greater side effects and complications. Future inventions should analyze compliance in home-based in-person interventions. 

### 4.3. Assessment and Intervention 

One of the strengths of this study is that intervention was tailored based on pre-assessment, taking into account the oncology process, physical status and capacity, as well as patients’ interests and preferences, following current literature in the oncology field [22,23,24,25,29,32]. This allowed patients to be stratified in groups based on physical activity level and functional status to prioritize intervention type. The FIIT (Frequency, Intensity, Time and Type) formula was also followed [7]. Furthermore, training principles were further considered [7] and explained in the present manuscript (Appendix A). Concerning the strengths of this study, it should be noted that our intervention was safe, and no adverse events were reported in the study participants. 

Pre-assessment allowed a supervised, tailored prospective design: while the physical and functional assessment paid attention to the patient’s physical status, the clinical interview allowed personal interests, needs and preferences to be taken into account. A patient-centred intervention is considered the best clinical practice in exercise intervention in women with advanced BC [28].

There is limited information evaluating the prognostic value of a TE intervention in women with MBC, in contrast to the growing work in early BC that reports a decreased risk of cancer-related and overall mortality. One study analyzed secondary data of a psychotherapy clinical trial with more than 100 MBC patients and showed a benefit in survival, with the major limitation of relying on self-reported questionnaires to draw these conclusions [13]. 

The intensity of the present TE program was moderate, consisting of aerobic exercise at 60–80% age-predicted maximum HR and resistance exercise at 70% of estimated 1-RM. This matches the combined exercise type intervention, with aerobic training ranging from 55–85% maximum HR and resistance exercise ranging from 40–90% of RM [28]. Apart from the similarities, this intervention was modified as needed in order to guarantee safety (Appendix A). 

### 4.4. Patient-Reported and Functional Outcomes 

Regarding functional outcomes, handgrip strength at baseline ranged from 3.33 to 29.33 kg. A group of 71 MBC had a mean value of 26.6 (6.0) kg, significantly lower than their matched controls [45]. Lower values in handgrip strength are associated with physical frailty and are predictive of disability in older people [46]. For example, a cut-off point of 17.4 kg identifies patients with mobility limitations in older women [47]. 

The number of repetitions performed during 30-STS ranged from one single repetition to 26 repetitions at baseline levels, while 2 out of 30 patients were not able to perform the test. Regarding 30-STS, older metastasis patients (mean age = 62.6 years) have shown lower levels of 30-STS repetitions (11.6 [0.38]) [42] when compared to matched controls (22 [7]) [48]. In the group that completed the TE program (*n* = 11), 30-STS repetitions increased from 14.50 to 19.61. In patients with spinal metastases, a TE program involving isometric spinal muscle strengthening increased 30-sts values from 5.1 (1.4) to 9.0 (2.6) [49]. Increasing lower limb strength is vital, as 30-STS repetitions below 15 predicts risks of falls and fractures in older healthy patients [50].

It should be highlighted that the level of physical activity (IPAQ) from the left or not started group (2.351 METS) and the compliance < 75% group (3.583 METS) was lower than the group with higher compliance (6.675) measured by IPAQ–SF (Table 3). A study with a sample of 85 patients with bone metastases (45 of them with breast cancer) found an inverse relationship between the level of physical activity and outcomes such as pain score or perceived physical function [51]. 

### 4.5. Improvements in Outcomes

Patients who completed the program (*n* = 11) decreased their CRF and improved their physical function measured by handgrip strength and functional tests such as 30-STS, lie-to-sit transition and adapted burpees. However, the lack of a control group did not allow comparison. The heterogeneity was also a limitation, as women undergoing various forms of systemic therapy and who were at different points during their metastatic disease were included. This contributed to the variation in functional parameters. However, findings in terms of low compliance and sample heterogeneity represent the implementation of TE in the real-world setting. 

## 5. Conclusions

This study showed that an individualized, supervised TE program is safe and feasible in MBC, although with low compliance due to personal matters and tumor/treatment-related issues. Patients who completed the program decreased their CRF and improved their physical function measured by handgrip strength and functional tests such as 30-STS, lie-to-sit transition and adapted burpees. Given the heterogeneity in the clinical status of patients and the degree of compliance, future research should include a wider sample to further analyze the effects of TE programs in this population. 

## Figures and Tables

**Figure 1 ijerph-19-11203-f001:**
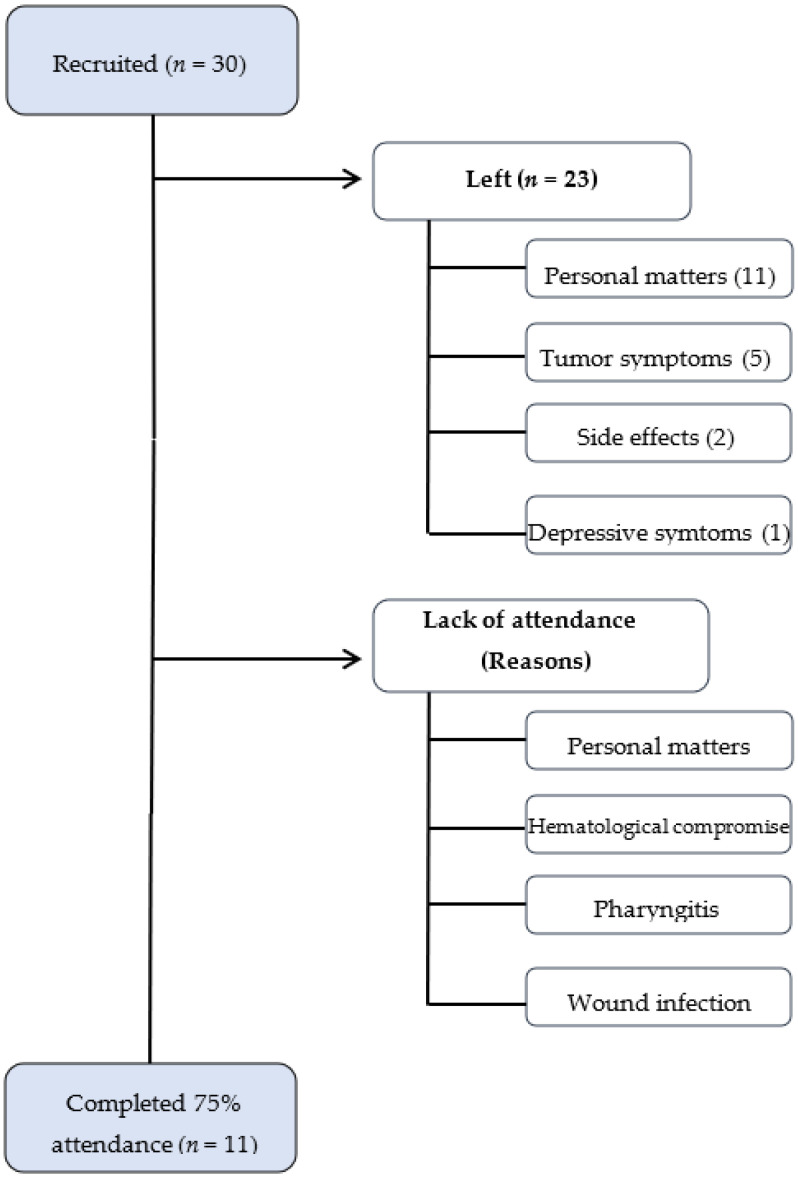
Flowchart from compliance.

**Table 1 ijerph-19-11203-t001:** Patients’ clinical variables at baseline.

	Baseline Group (*n* = 30)	Intervention Group: Compliance > 75% Group (*n* = 11)
Age (years)—Mean (SD)	52.46 (8.27)	52 (10.27)
Weight (kg, SD)	71.2 (8.2)	68.7 (6.8)
Height (cm, SD)	160.3 (12.3)	158.8 (16.1)
Body mass index (kg/m^2^, SD)	27.0 (2.3)	27.1 (3.1)
**Affected breast side *n* (%)**		
Right	11 (36.7%)	4 (36.4%)
Left	15 (50%)	7 (63.6%)
Bilateral	4 (13.3%)	0
Lymphedema	7 (23.3%)	3 (27.3%)
**Comorbidities/CV risk factors**		
Arterial hypertension	3 (10%)	1 (9.1%)
Diabetes	1 (3.3%)	0
Hyperlipemia	4 (13.3%)	1 (9.1%)
Smoker	0	0
Ex-smoker	10 (3.3%)	4 (36.4%)

**Table 2 ijerph-19-11203-t002:** Patients’ oncological characteristics at baseline.

	Baseline Group (*n* = 30)	Intervention Group: Compliance > 75% Group (*n* = 11)
Hystologic subtype		
HHRR positive—HER2 neg	21 (70%)	8 (73%)
HHRR positive—HER2 pos	3 (10%)	2 (18%)
Triple-negative	3 (10%)	1 (9%)
HHRR neg—HER2 positive	3 (10%)	0 (0%)
Type of surgery		
Mastectomy	21 (70%)	8 (72.7%)
Breast-conserving	6 (20%)	3 (27.3%)
None	3 (10%)	-
Line of treatment		
1st	20 (67%)	8 (73%)
2nd	6 (20%)	2 (18%)
3rd	4 (13%)	1 (9%)
Type of systemic treatment		
Chemotherapy	9 (30%)	3 (27.3%)
ET	8 (27%)	4 (36.4%)
CT + monoclonal ab	2 (7%)	0
Monoclonal ab	3 (10%)	1 (9.1%)
ET ^1^ +/− CDK inhib	8 (26%)	3 (27.3%)
Site of metastatic disease		
Visceral (liver, lung or CNS)	6 (20%)	3 (27 %)
Non-visceral	15 (50%)	3 (27 %)
Visceral and Non-visceral	9 (30%)	5 (46 %)
Bone metastasis	22 (73%)	8 (73%)
Spine	17 (57%)	5 (62%)
Pelvis	11 (36%)	4 (50%)
Thorax	12 (40%)	4 (50%)
Femur	8 (26%)	3 (37%)
Number of metastases		
Oligometastasis (1–3)	5 (17%)	3 (27%)
Multiple metastasis	25 (83%)	5 (73%)
Type of bone metastases		
Mixed	7 (23%)	2 (25%)
Osteoblast	8 (27%)	4 (50%)
Osteolytic	7 (23%)	2 (25%)

^1^ ET: endocrine therapy, CT: chemotherapy, CDK inhib: cyclin-dependent kinase inhibitors.

**Table 3 ijerph-19-11203-t003:** Patients’ baseline functional and self-reported outcomes in each attendance group.

	Left or Not Started Group (*n* = 11)	Compliance < 75% Group (*n* = 8)	Intervention Group: Compliance > 75% Group (*n* = 11)
Handgrip strength (kg)	20.09 (5.63)	21.74 (7)	19.06 (8.32)
30-STS (*n*)	14 (6.28)	15.78 (5.36)	14.50 (4.94)
Lie-to-sit	7 (2.30)	5 (1)	7.87 (3.92)
Adapted burpees (*n*)	-	53	91
CRF (0–10)	5.08 (2.86)	6.15 (2.26)	5.54 (3.37)
ULFI (0–100)	63.4 (19.18)	59 (39.67)	64.72 (19.12)
LLFI (0–100)	55.60 (28.99)	60.85 (30.95)	60.18 (29.23)
EORTC QLQ–C30	67.10 (13.71)	60.14 (11.83)	60.18 (11.99)
EORTC QLQ–BR23	41.77 (9.94)	42.28 (7.43)	49 (10.14)
IPAQ–SF (METS)	2.351 (1.825)	3.583 (1.550)	6.675 (8.492)

30-STS: 30 s sit-to-stand test, CRF: cancer-related fatigue, ULFI: Upper Limb Functional Index, LLFI: Lower Limb Functional Index, IPAQ-SF: International Physical Activity Questionnaire–Short Form, QoL: Quality of Life, QLQ–C30: the European Organization for Research and Treatment of Cancer Quality of Life Questionnaire Core 30, QLQ–BR23: the European Organization for Research and Treatment of Cancer Breast Cancer–Specific Quality of Life Questionnaire.

**Table 4 ijerph-19-11203-t004:** Patients’ functional and self-reported outcomes from intervention group (*n* = 11).

	Pre-Intervention	Post-Intervention	Mean Change
Handgrip strength (kg)	19.06 (8.32)	19.16 (6.80)	0.1
30-STS (*n*)	14.50 (4.94)	19.61 (6.27)	5.11
Lie-to-sit	7.87 (3.92)	8 (1.32)	0.13
Adapted burpees (*n* = 1)	91	101	10
CRF (0–10)	5.54 (3.37)	4.33 (1.86)	−1.21
ULFI (0–100)	64.72 (19.12)	60.18 (17.49)	−4.54
LLFI (0–100)	60.18 (29.23)	56.90 (27.87)	−3.28
EORTC QLQ–C30	60.18 (11.99)	61.72 (13.52)	1.54
EORTC QLQ–BR23	49 (10.14)	42.81 (8.49)	−6.19
IPAQ–SF	6.675 (8.492)	6.746 (5.148)	71

CRF: cancer-related fatigue, ULFI: Upper Limb Functional Index, LLFI: Lower Limb Functional Index, IPAQ–SF: the International Physical Activity Questionnaire–Short Form, QoL Quality of Life, QLQ–C30: the European Organization for Research and Treatment of Cancer Quality of Life Questionnaire Core 30, QLQ–BR23: the European Organization for Research and Treatment of Cancer Breast Cancer–Specific Quality of Life Questionnaire; 30-STS: 30 s sit-to-stand test.

## Data Availability

The datasets analyzed during the current study are available from the corresponding author on reasonable request.

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
