# Peer review of "Implementation of a Standard Care Program of Therapeutic Exercise in Metastatic Breast Cancer Patients"

_ijerph, 2022, doi:10.3390/ijerph191811203_

Round 1
Reviewer 1 Report
Pajares et al should be congradulated for the manuscript. Although they have a small population, there is little evidence on this topic.
Minor suggestions:
-the authors should include the left ventricle function (at least preserved / not preserved) of the patients
- Page 8: Lack of attendance ("Reasons") - adjust
Author Response
Implementation of a Standard Care Programme of Therapeutic Exercise in Metastatic Breast Cancer Patients
Minor revision. Response to reviewer 1.
Comments and Suggestions for Authors
Pajares et al should be congradulated for the manuscript. Although they have a small population, there is little evidence on this topic.
Authors: Thank you so much.
Minor suggestions:
-the authors should include the left ventricle function (at least preserved / not preserved) of the patients
Authors: Thank you so much.
We agree that this data is very important, especially in the field of cardio-oncology. However, after making the necessary revisions, we were unable to access that data.
- Page 8: Lack of attendance ("Reasons") – adjust
Authors: Thank you. It has been fixed.

Reviewer 2 Report
This study assessed the feasibility and potential benefit of therapeutic exercise (TE) intervention in women with metastatic breast cancer (MBC) and reported that the TE intervention was safe and feasible in women with MBC and improved cancer-related fatigue. Although the TE intervention has a very low compliance rate (36%), it is one of the challenging issues motivating women with MBC to do regular physical exercise in the real clinical environment.
Comment on study
In the abstract, result, and discussion section, the authors concluded that TE intervention improved fatigue and increased participants’ functional parameters. However, I could not find statistical analysis data (pre-post intervention’s p-value or effect size) in Table 4. So not sure, the study found a trend of improvement or statistically significant improvement.
Minor issues,
It may not require subtitles (Recruitment, Compliance, Assessment and Intervention, Patient-reported and functional outcomes, Improvements in outcomes) in the discussion section.
In the discussion section, it will be better to elaborate on the important findings of the current study, interpretation of results compared with similar published studies in this area, overall significance and weaknesses of the study, and recommendations for future researchers.
Author Response
Implementation of a Standard Care Programme of Therapeutic Exercise in Metastatic Breast Cancer Patients
Minor revision. Response to reviewer 2.
Comments and Suggestions for Authors
This study assessed the feasibility and potential benefit of therapeutic exercise (TE) intervention in women with metastatic breast cancer (MBC) and reported that the TE intervention was safe and feasible in women with MBC and improved cancer-related fatigue. Although the TE intervention has a very low compliance rate (36%), it is one of the challenging issues motivating women with MBC to do regular physical exercise in the real clinical environment.
Authors: Thank you for your comment. In fact, authors think that the low compliance rate obtained should be shared with clinicians and researchers to highlight the challenge of exercise prescription in this population.
Comment on study
In the abstract, result, and discussion section, the authors concluded that TE intervention improved fatigue and increased participants’ functional parameters. However, I could not find statistical analysis data (pre-post intervention’s p-value or effect size) in Table 4. So not sure, the study found a trend of improvement or statistically significant improvement.
Authors: Thank you. Results are a trend. As you can see in the statistical analysis section, we only performed mean and standard deviation. For this reason, we did not write anything related to significance or even mean differences. In this manuscript, authors aimed to report the results about this intervention in the clinical field.
For this reason and trying to be as clear as possible, in the results there is a section for descriptive variables (“Patient-reported and functional outcomes at baseline”) and another subheading for improvements in variables “Improvements in outcomes”). In Table 4, "pre-intervention" and "post-intervention" values can be observed, but with no difference in means or any type of statistical significance.
However, reviewer #1 requested for basic arithmetic value for increase or decrease, so it was added in table 4 (see new column in tracked changes), and Statistical analysis section was modified accordingly
Minor issues,
It may not require subtitles (Recruitment, Compliance, Assessment and Intervention, Patient-reported and functional outcomes, Improvements in outcomes) in the discussion section.
Authors: Thank you. We would like to keep those subheading, especially so that readers can see the difference between the descriptive variables at baseline (“Patient-reported and functional outcomes at baseline”) and differences between baseline outcomes and post-intervention outcomes (“Improvements in outcomes”). In this way, the results on "compliance", which are specific to the implementation model, are also differentiated.
In the discussion section, it will be better to elaborate on the important findings of the current study, interpretation of results compared with similar published studies in this area, overall significance and weaknesses of the study, and recommendations for future researchers.
Authors: Thank you. Regarding significance and important findings, we added “ Therefore, the results of this study may be of interest to clinicians and researchers when implementing and designing TEP in MBC patients in a real-world setting.”. On the other hand, the weakness and recommendations for future research are already mentioned though the discussion section (i.e “One strength of the present study was that oncologists recruited patients interested in the intervention “; “The main obstacle in developing our TE program was drop-out” and “future research should include a wider sample to further analyze the effects of TE programs in this population”.
With regards with comparison with other studies, this is limited due to the lack of research on this population. However, we have added new information related to descriptive data, as follows:
“Lately, it should be highlighted that the level of physical activity (IPAQ) from the left or not started group (2.351 METS) and the compliance <75% group (3.583 METS) was lower than the group with higher compliance (6.675) measured by IPAQ-SF (table 3). A study with a sample of 85 patients with bone metastases (45 of them with breast cancer) found an inverse relationship between the level of physical activity and outcomes such as pain score or perceived physical function [53]. “
New reference:
- Guinan, E.M.; Devenney, K.; Quinn, C.; Sheill, G.; Eochagáin, C.M.; Kennedy, M.J.; McDermott, R.; Balding, L. Associations Among Physical Activity, Skeletal Related Events, and Patient Reported Outcomes in Patients with Bone Metastases. Semin. Oncol. Nurs. 2022, 38, 151274, doi:10.1016/j.soncn.2022.151274.

Reviewer 3 Report
Physical therapy for cancer patients is very important. Therefore, it is considered a clinically meaningful study. However, I have a question about something I don't understand.
1) What is the difference between TEP and TEEP?
2) Did you use a formula such as G-power for 30 subjects?
3) Please provide the basic arithmetic value for increase or decrease in each table.
4) Why were there so many dropouts in the experimental group?
5) Please provide the rationale for the treatment method used in this study.
6) The title of Table 4 needs to be corrected. (Patient', Patient'......)
Author Response
Implementation of a Standard Care Programme of Therapeutic Exercise in Metastatic Breast Cancer Patients
Minor revision. Response to reviewer 3.
Comments and Suggestions for Authors
Physical therapy for cancer patients is very important. Therefore, it is considered a clinically meaningful study. However, I have a question about something I don't understand.
1) What is the difference between TEP and TEEP?
Authors: Thank you. Those definitions are explained in the introduction section:
- TE= therapeutic exercise, lines 43-44
- TEP=therapeutic exercise programme, lines 71-72.
- TEEP was a typing mistake, so it has already been fixed in lines 82 and 85, thank you.
2) Did you use a formula such as G-power for 30 subjects?
Authors: Thank you. Sample size was not calculated, as this is an implementation study and, from our point of view, one of the highlights of the present study are data related to low compliance rate and drop outs. The sample is insufficient for in-depth statistical analysis, and this limitation is reported along the manuscript.
3) Please provide the basic arithmetic value for increase or decrease in each table.
Authors: Thank you. It has been added, and statistical analysis section has been modified accordingly.
4) Why were there so many dropouts in the experimental group?
Authors: Thank you. This is explained in “compliance” subheading from results section, as follows:
“19 patients abandoned the program due to different reasons: 11 patients dropped out because of personal matters (transport, family problems, distance to hospital, lack of motivation); 5 patients presented tumor symptoms (3 bone pain, 1 liver pain, and 1 thrombosis) that prevented them from attending the sessions; 2 patients had serious treatment side effects; and 1 patient presented major depressive symptoms that made it impossible to complete the program.
11 patients completed the program, representing 36% compliance. Lack of attendance was due to personal matters and health status, namely hematological compromise (absolute neutrophil counts), pharyngitis, and wound infection. 4/11 lacked some functional parameters because they were unable to attend on the day of the pre- or post-evaluation. “
5) Please provide the rationale for the treatment method used in this study.
Authors: Thank you. The rationale is fully described in the Appendix. For example, the main adaptation targeted was decided by the physical therapist and described in the table. Neuromuscular and cardiorespiratory adaptations, the exercise prescription is also detailed in the Appendix, including the references and research it has been based on.
6) The title of Table 4 needs to be corrected. (Patient', Patient'......)
Authors: Thank you so much. It has been corrected.
